# An Innovative Structural Rearrangement in Imine Palladacycle Metaloligand Chemistry: From Single-Nuclear to Double-Nuclear Pseudo-Pentacoordinated Complexes

**DOI:** 10.3390/molecules28052328

**Published:** 2023-03-02

**Authors:** Basma al Janabi, Francisco Reigosa, Gemma Alberdi, Juan M. Ortigueira, José M. Vila

**Affiliations:** Department of Inorganic Chemistry, University of Santiago de Compostela, E-15702 Santiago de Compostela, Spain

**Keywords:** palladacycles, imines, catalysis, Suzuki, cross coupling

## Abstract

Treatment of the double nuclear complex **1a**, di-μ-cloro-bis[N-(4-formylbenzylidene)cyclohexylaminato-C6, N]dipalladium, with Ph_2_PCH_2_CH_2_)_2_PPh (triphos) and NH_4_PF_6_ gave the single nuclear species **2a**, 1-N-(cyclohexylamine)-4-N-(formyl)palladium(triphos)(hexafluorophasphate). Reaction of **2a** with Ph_2_PCH_2_CH_2_NH_2_ in refluxing chloroform via a condensation reaction of the amine and formyl groups to produce the C=N double bond, gave **3a**, 1-N-(cyclohexylamine)-4- N-(diphenylphosphinoethylamine)palladium(triphos)(hexafluorophasphate); a potentially bidentate [*N*,*P*] metaloligand. However, attempts to coordinate a second metal by treatment of 3a with [PdCl_2_(PhCN)_2_] were to no avail. Notwithstanding, complexes **2a** and **3a** left to stand in solution spontaneously self-transformed to give in either case the double nuclear complex **10**, 1,4-N,N-terephthalylidene(cyclohexilamine)-3,6-[bispalladium(triphos)]di(hexafluorophosphate), after undergoing further metalation of the phenyl ring, then bearing two mutually *trans* [Pd(Ph_2_PCH_2_CH_2_)_2_PPh)-*P*,*P*,*P*] moieties: an unprecedented and serendipitous result indeed. On the other hand, reaction of the double nuclear complex 1b, di-μ-cloro-bis[N-(3-formylbenzylidene)cyclohexylaminato-C6, N]dipalladium, with Ph_2_PCH_2_CH_2_)_2_PPh (triphos) and NH_4_PF_6_ gave the single nuclear species **2b**, 1-N-(cyclohexylamine)-4-N-(formyl)palladium(triphos)(hexafluorophasphate), Treatment of **2b** with H_2_O/glacial MeCOOH gave cleavage of the C=N double bond and of the Pd···N interaction, yielding **5b**, isophthalaldehyde-6-palladium(triphos)hexafluorophosphate, which then reacted with Ph_2_P(CH_2_)_3_NH_2_ to yield complex **6b**, N,N-(isophthalylidene(diphenylphosphinopropylamine)-6-(palladiumtriphos)di(hexafluorophosphate), with two pairs of non-coordinated nitrogen and phosphorus donor atoms. Treatment of **6b** with [PdCl_2_(PhCN)_2_], [PtCl_2_(PhCN)_2_], or [PtMe_2_(COD)] gave the new double nuclear complexes **7b**, **8b** and **9b**, palladiumdichloro-, platinumdichloro- and platinumdimethyl[N,N-(isophthalylidene(diphenylphosphinopropylamine)-6-(palladiumtriphos)(hexafluorophosphate)-P,P], respectively, showing the behavior of **6b** as a palladated bidentate [*P*,*P*] metaloligand. The complexes were fully characterized by microanalysis, IR, ^1^H, and ^31^P NMR spectroscopies, as appropriate. The X-ray single-crystal analyses for compounds 10 and 5b have been previously described as the perchlorate salts by JM Vila et al.

## 1. Introduction

The organometallics known as cyclometallated complexes [1,2], with special emphasis focused on the palladium derivatives, i.e., the palladacycles [3,4,5], have developed over the years into a large group of species. Their structural-related characteristics, reactivity patterns and most of all their applications have brought them to the highest echelon of research. It is noteworthy to mention their usage as metalomesogens [6] as antineoplastic substances [7,8] and in synthetic chemistry, where they have shown a paramount role as catalysts and pre-catalysts [9,10,11,12,13,14] inclusive of the especial case of the phosphorus donor ligands designed by Hermann et al. [15,16] and of the pioneering work by Suzuki [17,18]. As far as their reactivity is concerned, it derives fundamentally from the different nature of the bonds on the metal atom, i.e., the M-C and M-Y (Y = donor atom) bonds, which allow a great variety of processes ranging from insertions [19] to reactions with a rather large number of nucleophiles [20,21], offering a plentiful and alluring chemistry. In the case of palladacycles derived from imines the stereochemistry of the four-coordinated palladium(II) atom is the characteristic square-planar geometry; notwithstanding, five- and six-coordinate palladium complexes have also been reported [22,23,24]. Precisely because the preferred geometry for palladium is square-planar, one would expect that with tridentate polyphosphanes, namely (Ph_2_PCH_2_CH_2_)_2_PPh (triphos), an uncoordinated phosphorus atom to remain upon binding of the phosphane to palladium. Whereupon they should then be suitable for coordination to another metal and give way to new heterobimetallic compounds. However, this is not the case as we have previously described in the reaction of triphos with dinuclear (μ-Cl)_2_-bridged palladacycles containing imine, bis(imine) and the related azobenzene ligands. In the resulting complexes, all three phosphorus donors are bonded to the metal to give complexes in which the palladium-nitrogen bond distance; though slightly longer than the value expected for a true Pd-N bond 2.23(4) Å, this suggests pentacoordination of the metal atom, with the latter in a distorted trigonal bipyramid geometry, as depicted in Figure 1.

Thus, what began as a fortuitous finding resulted in a systematic behavior of these palladacycles was then extended to other organic metalated moieties [25]. More recently, examples of related five-coordinate palladium complexes have appeared with triphosphanes [26], triptycene-based ligands, an example of how especially designed ligands may induce five-coordination [27], and in oxime palladacycles with simultaneous bonding of m-dppe and dppe-*P*,*P* to the palladium center [28]. Yet, the story does not end here, since in our efforts to study new patterns of reactivity in single-nuclear palladacycles containing triphos, we have found that a double-nuclear complex is serendipitously obtained; in fact, two different complexes gave the same structure after recrystallization. This is a more than surprising result that has led us to change the initial course of this manuscript by focusing our efforts on the search for a satisfactory explanation, as well as on the related chemistry of the initial palladacycles themselves. Herein, we report the route leading to this conversion along with a tentative explanation for this chemistry.

## 2. Results

For the convenience of the reader, the compounds and reactions are shown in Figure 2 and Figure 3. The compounds described in this paper were characterized by elemental analysis (C,H,N) and by IR and ^1^H, ^31^P–{^1^H} NMR spectroscopy (data in Section 3), as well as by X-ray diffractometric analysis (**5b**, **10**). The cyclometalated complexes **1a** [10], **1b** [11], **2a**, **2b** [12] have been reported previously; **2a**, **2b** as the perchlorate salts.

The aim of this paper was to show that palladacycles bearing an uncoordinated formyl group could be conveniently transformed into either bidentate-[P,N] or bidentate-[P,P] palladium metaloligands. The first step would be to hinder further reactions of the palladium center, for which purpose the triphos derivatives were chosen; this tightly holds the metal by simultaneous bonding to the metallated phenyl carbon and to the three phosphorus donors of the tridentate phosphane ligand. There are then two possibilities: the first is the addition of new [P,N] donors by direct restoration of the C=N bond with a convenient aminophosphane; the second one is cleavage of the C=N double bond within the metallacycle ring, to give a palladium complex with two formyl moieties, and the addition of two sets of [P,N] donor atoms via the formation of two new C=N bonds. In the latter case, one could expect bis(bidentate-[P,N]) or bidentate-[P,P] coordination, the outcome depending on the length of the aminophosphane ligand. This was achieved (in part) as we describe, vide infra, together with a new and most striking result that somewhat changed our initial research pathway. Thus, treatment of precursor **1a** with Ph_2_PCH_2_CH_2_)_2_PPh (triphos) and NH_4_PF_6_ produced the single nuclear complex **2a**, which was fully characterized (see Experimental). Then, treatment of complex **2a** with Ph_2_PCH_2_CH_2_NH_2_ in boiling chloroform for 8 h gave **3a**. The IR spectrum showed absence of the υ(C=O) stretch and two υ(C=N) bands at 16,371,623 cm^−1^; the second corresponds to the C=N moiety bonded to the metal atom [29,30]. The ^1^H NMR spectrum showed two singlet resonances at 8.21, 8.13 ppm; the first (higher frequency) was assigned to the free HC=N proton and the second (lower frequency) to the HC=N proton of the cyclometallated ring [31]. An apparent triplet resonance at 5.83 was assigned to the H5 proton showing coupling to the phosphorus nucleus *trans* to carbon with ^4^*J*(HH) ≈ ^4^*J*(PH). The new metaloligand was reacted with PdCl_2_ or PtCl_2_ in CHCl_3_ at room temperature in the hope of producing complexes of type **4a**, but to no avail. However, a solution of **2a**, on the one hand, and the subsequent solution of **3a**, on the other, left to stand for recrystallization by slow evaporation of a chloroform solution gave an unprecedented and serendipitous result which in both cases produced crystallization of the same PF_6_ salt of a double-nuclear palladacycle, **10**; a structure altogether distinct from the one expected on the basis of NMR data. This modified our initial objectives, prompting us to seek a plausible explanation for this finding, and left the pursuit of yet new metaloligands for the last part of the manuscript (vide infra).

Likewise, reaction of the precursor **1b** with Ph_2_PCH_2_CH_2_)_2_PPh (triphos) and NH_4_PF_6_ produced the single nuclear complex **2b**, which was fully characterized (see Experimental). Treatment of **2b** with a mixture of acetic acid/water gave **5b** (see Experimental).

A crystal structural analysis for complex **5b** could be performed. Thus, suitable crystals of **5b** were grown by slowly evaporating a chloroform solution of the compound. Crystal data are given in the Supporting Information. The ORTEP illustration of complex **5b** is shown in Figure 1. The compound crystalizes in the P2_1_/n space group with two acetone solvent molecules. The crystals consist of discrete molecules separated by van der Waals distances.

The four coordinated palladium atom is bonded to the phenyl carbon atom of the metallated phenyl ring, and to three phosphorus atoms of the triphos lignad. The coordination sphere of the palladium may be described as slightly distorted square-planar; the palladium atom is 0.136 from the mean coordination plane [C(6), P(1), P(2), P(3)]. The angles between adjacent atoms in the coordination sphere of palladium are fairly close to the expected value of 90° with the smaller values for the bite angles for the tridentate ligand [P(1)-Pd(1)-P(2) = 83.97° and P(2)-Pd(1)-P(3) = 83.48°] consequent upon chelation and the sum of angles at palladium is 359.44°; the palladium coordination plane [Pd(1), C(6), P(1), P(2), P(3)] and the metalated phenyl ring are at 76.16°. The C(2)-C(7)-O(1) oxygen is directed toward the palladium atom; however, the Pd(1)-O(1) distance of 2.780 Å precluded any interaction between both atoms.

Treatment of **5b** with two equivalents of 3-diphenylphosphinopropylamine, Ph_2_PCH_2_CH_2_CH_2_NH_2_, gave **6b** as an air-stable solid (see Experimental): a novel tetradentate [N,P,N,P] metaloligand. The IR and NMR spectra showed the absence of the υ(C=O) stretches and of the *H*C=O resonances, respectively. Instead, two bands at 1639 and 1625 cm^−1^ were assigned to the υ(C=N) vibrations in the IR spectrum. The ^1^H NMR spectrum showed two resonances at 8.03 and 7.97 ppm ascribed to the two non-equivalent *H*C=N protons. Several coordination possibilities arise, considering there are four new donor atoms; however, attempts to prepare complexes inducing [N,P,N,P] tetracoordination or alternatively bis(bidentate-[P,N]) coordination failed, and only compounds with **6b** as bidentate-[P,P] metaloligand could be achieved. Thus, reaction of **6b** with [PdCl_2_(PhCN)_2_], [PtCl_2_(PhCN)_2_] or [PtMe_2_(COD)] gave the new dinuclear complexes **7b**-**9b**, as air-stable solids which were fully characterized (see Experimental). The most salient feature of the NMR spectra of **7b**–**9b** as opposed to **6b** was the downfield shift of the ^31^P resonance for the non-coordinated PPh_2_ donors from −18.3 (**6b**) ppm to 17.9 (**7b**), −4.9 (**8b**) and 18.0 (**9b**). The latter two also showed the phosphorus coupling to the platinum nucleus, [J(PtP) = 3296] and [J(PtP) = 2426], respectively.

The molecular structure of **10** contains a discrete centrosymmetric dinuclear cation of [{(Ph_2_PCH_2_CH_2_)_2_PPh-P,P,P}Pd{N(Cy) = (H)C}C_6_H_2_{C(H) = N(Cy)}Pd{(Ph_2_PCH_2_-CH_2_)_2-_PPh-P,P,P}]^2+^ (half of the cation per asymmetric unit) and two hexafluorophosphate ions (crystal data are given in the Supporting Information; the ORTEP illustration of complex **10** is shown in Figure 2). The cation is analogous to the one prepared and discussed by us earlier as the ClO_4_ salt [25]; therefore, we did not consider it necessary to discuss the structure again. Nevertheless, an important point should be considered here in relation to the Pd···N distance that marks the penta-coordination environment on the palladium atom namely 2.2869(19) (structure from compound **2a**) and 2.296 Å (structure from compound **3a**), being less than our previous finding of 2.338 Å [25], and it is, to the best of our knowledge, the shortest distance known for this type of compounds, and very close the palladium-nitrogen bond length found in an authentic pentacoordinated palladium(II) complex of 2.23(2) Å [32]. To seek a plausible explanation for this, let us say irrefutable crystallographic results, we reasoned that formation of complex **10** requires activation of a second phenyl C-H bond, with metallation by a Pd(triphos) moiety and condensation of a cyclohexylamine to the formyl group; additionally, it would be expected that the presence of trace amounts of acid and/or water should facilitate the process. Likewise, to check that this is not a one-time reaction, compound **2a** was left to stand in CDCl_3_ in an NMR tube, whereupon resonances other than those expected seemed to slowly dawn within the first 24 h advocating evolution of a novel compound. Then, the CDCl_3_ NMR solution was left to stand further upon which single-crystals appeared; an X-ray crystallographic analysis showed them to be compound **10**. Initially, the ^1^H NMR spectra showed a mixture of two products assignable to complex **2a** and an emerging compound **10**. Thus, resonances of circa 9.0, 8.5 and 6.2 ppm were assigned to the *H*C=O, *H*C=N and H5 protons, respectively, for **2a** (Figure 3, red circles), and new resonances of circa 8.2 and 5.7 ppm were assigned to the *H*C=N and H5 protons, respectively, for **10** (Figure 3, blue circles).

Additionally, the ^31^P{H} NMR spectrum also showed triplet and doublet resonances assignable to compound **2a**, circa 86 and 45 ppm, respectively (Figure 4, red circles), plus another triplet resonance at circa 82 ppm, and two doublets at circa 43 ppm (Figure 4, blue circles), which were assigned to the central ^31^P nucleus and to the two inequivalent phosphorous, respectively, for compound **10**. This transformation was even more apparent in time and intermediate species could also be detected, especially in the ^31^P NMR spectra and tentatively assigned (Figure 4). Therefore, following these experimental observations and considering that it is well known that commercial solvents, inclusive of NMR solvents, contain traces of acid and water, as is the case with CHCl_3_ and CDCl_3_, and most important of all the irrefutable evidence of crystal formation of the dinuclear palladium complex, we very tentatively propose the following mechanism for this transformation, an unprecedented reactivity pattern, Figure 4. The first step is hydrolysis of the C=N double bond in compound **2a** by trace acid/water producing the diformyl palladated species **2a_2_** and cyclohexylamine; the latter may undergo condensation with the HC=O group in **2a** to give **2a_1_**. Species **2a_2_** undergoes de-metallation liberating terephthalaldehyde, **2a_3_**, and the electrophile Pd(triphos), which activates the phenyl C-H bond in **2a_1_** metallating the aromatic ring to yield the double-nuclear complex **10**. We also suggest that solvent coordination may possibly stabilize the intermediate species.

## 3. Experimental Methods

### 3.1. General Procedures

Solvents were purified by standard methods [33]. The reactions were carried out under dry nitrogen. The triphosphine (Ph_2_PCH_2_CH_2_)_2_PPh (triphos), 2-diphenyl-phosphinoethanoamine, Ph_2_P(CH_2_)_2_NH_2_ and 3-diphenylphosphinopropanoamine Ph_2_P(CH_2_)_3_NH_2_ were purchased from Aldrich-Chemie. PdCl_2_, PtCl_2_ and PtMe_2_(COD) were purchased from commercial sources. Elemental analyses were performed with a Thermo Finnigan elemental analysis, model Flash 1112. IR spectra were recorded on Jasco model FT/IR-4600 spectrophotometer. ^1^H NMR and spectra in solution were recorded in acetone−d6 or CDCl3 at room temperature on Varian Inova 400 spectrometers operating at 400 MHz using 5 mm o.d. tubes; chemical shifts, in ppm, are reported downfield relative to TMS using the solvent signal as reference (acetone−d_6_ δ ^1^H: 2.05 ppm, CDCl_3_ δ ^1^H: 7.26 ppm). ^31^P NMR spectra in solution were recorded in acetone−d_6_ or CDCl_3_ at room temperature on Varian Inova 400 spectrometer operating at 162 MHz using 5 mm o.d. tubes and are reported in ppm relative to external H_3_PO_4_ (85%). Coupling constants are reported in Hz. All chemical shifts are reported downfield from standards. The preparation of the cyclometalated complexes **1a** [10], **1b** [11], **2a**, **2b** [12] has been reported previously; **2a**, **2b** as the perchlorate salts.

### 3.2. Preparation of the Complexes

#### 3.2.1. Synthesis of **3a**

Compound **2a** (55 mg, 0.056 mmol) and 2-diphenylphosphinoethanoamine (14.90 mg, 0.065 mmol) were refluxed in chloroform (20 cm^3^) under argon for 3 h in a Dean–Stark equipment. After cooling to room temperature, the resulting solution was reduced to low volume. Then, after addition of n-hexane, the solid formed was filtered off and dried. Yield: 56.2 mg, 85%. Anal. found: C, 63.0; H, 5.3; N, 2.4. C_62_H_63_F_6_N_2_P_4_Pd requires C, 63.1; H, 5.4; N, 2.4%. IR: u(C=N) 1625m,1603m cm^−1^. ^1^H NMR (CDCl3, δ ppm, J Hz): 8.12 [s, 1H, HC=N], 6.81 [d, 1H, H2, J(PHH) = 7.3], 6.7 [d, 1H, H3, J(HH) = 7.3;] 6.04 [d, 1H, H5, J(PH) = 7.3]. ^31^P-{^1^H} NMR (CDCl_3_, δ ppm) 83.5 [t, 1P, J(PP) = 25.0 Hz], 42.7 [d, 2P].

#### 3.2.2. Synthesis of **5b**

Compound **2b** (55 mg, 0.056 mmol) and water (10 cm^3^) were added in glacial acetic acid (15 cm^3^) and the resulting mixture was stirred for 48 h at 25 °C. Then, the acetic acid was removed under reduced pressure and extracted with dichloromethane. After drying with anhydrous sodium sulfate, filtering and reducing the solvent volume, a yellow solid was obtained, which was recrystallized from dichloromethane/hexane. Yield: 42.20 mg, 82%. Anal. found: C, 54.7; H, 4.1. C_42_H_38_F_6_O_2_P_4_Pd requires C, 54.9; H, 4.2. IR: u(C=O) 1683s, 1659 s cm^−1^. ^1^H NMR (CDCl_3_, δ ppm, J Hz): 9.83, 9.66 [s, 2H, HC=O], 7.79[s, 1H, H2], 6.96[d, 1H, H4, J(HH) = 7.0], 6.54[dd, 1H, H5, J(PH) = 6.6]. ^31^P-{^1^H} NMR (CDCl_3_, δ ppm) 82.7 [t, 1P, J(PP) = 24.8 Hz], 41.6 [d, 2P].

#### 3.2.3. Synthesis of **6b**

Compound **6b** was synthesized from **5b** (40 mg, 0.044 mmol) and 2-diphenyl-phosphinopropanoamine (21.41 mg, 0.088 mmol) following a similar procedure to that for **3a** but using 1:2 molar ratio. Yield: 43.15 mg, 75%. Anal. found: C, 66.0; H, 5.3; N, 2.4. C_72_H_70_F_6_N_2_P_4_Pd requires C, 66.1; H, 5.4; N, 2.4%. IR: u(C=N) 1639 m,1625 m cm^−1^. ^1^H NMR (CDCl3, δ ppm, J Hz): 8.03, 7.97 [s, 2H, HC=N], 7.0 [s, 1H, H2], 6.7 [d, 1H, H4, J(HH) = 7.5] 6.04 [dd, 1H, H5, J(PH) = 7.4]. ^31^P-{^1^H} NMR (CDCl_3_, δ ppm) 89.0 [t, 1P, J(PP) = 25.0 Hz], 41.7 [d, 2P], -18,3 [s, 2P].

#### 3.2.4. Synthesis of **7b**

Compound **6b** (40 mg, 0.031 mmol) and [PdCl_2_(PhCN)_2_] (11.89 mg, 0.031 mmol) were added in dry oxygen-free chloroform (15 cm^3^) and the resulting mixture stirred at 25 °C for 2 h. The solvent was removed under reduced pressure and the residue recrystallized form chloroma/hexane. Yield: 28.08 mg, 61%. Anal. found: C, 58.2; H, 4.6; N, 1.8. C_72_H_70_Cl_2_F_6_N_2_P_4_Pd_2_ requires C, 58.2; H, 4.7; N, 1.9%. IR: u(C=N) 1625m,1603 cm^−1^. ^1^H NMR (CDCl3, δ ppm, J Hz): 8.0, 7.93 [s, 2H, HC=N], 6.8 [s, 1H, H2], 6.3 [d, 1H, H4, J(HH) = 7.7] 6.01 [dd, 1H, H5, J(PH) = 7.4]. ^31^P-{^1^H} NMR (CDCl_3_, δ ppm) 109.4 [t, 1P, J(PP) = 23.0 Hz], 44.2 [d, 2P], 17.9 [s, 2P].

Compounds **8b** and **9b** were prepared analogously from [PdCl_2_PhCN)_2_] and [PtMe_2_COD)], respectively.

#### 3.2.5. Synthesis of **8b**

Compound **6b** (40 mg, 0.031 mmol); [PtCl_2_(PhCN)_2_] (14.63 mg, 0.031 mmol) C_72_H_70_Cl_2_F_6_N_2_P_4_PdPt requires C, 54.9; H, 4.; N, 1.8%. Yield: 32.68 mg, 67%. Anal. found: C, 58.2; H, 4.6; N, 1.8. C_72_H_70_Cl_2_F_6_N_2_P_4_Pd_2_ requires C, 58.2; H, 4.7; N, 1.9%. IR: u(C=N) 1625m,1603 cm^−1^. ^1^H NMR (CDCl3, δ ppm, J Hz): 8.0, 7.93 [s, 2H, HC = N], 6.8 [s, 1H, H2], 6.3 [d, 1H, H4, J(HH) = 7.7] 6.01 [dd, 1H, H5, J(PH) = 7.4]. ^31^P-{^1^H} NMR (CDCl_3_, δ ppm) 109.3 [t, 1P, J(PP) = 23.0 Hz], 44.1 [d, 2P], -4.9 [s, 2P, J(PtP) = 3296].

#### 3.2.6. Synthesis of **9b**

Compound **6b** (40 mg, 0.031 mmol); [PdMe_2_(COD)] (10.33 mg, 0.031 mmol) C_74_H_76_Cl_2_F_6_N_2_P_4_PdPt requires C, 5.4; H, 4.8; N, 1.7%. Yield: 35.3 mg, 71%. Anal. found: C, 58.2; H, 4.6; N, 1.8. C_72_H_70_Cl_2_F_6_N_2_P_4_Pd_2_ requires C, 58.2; H, 4.7; N, 1.9%. IR: u(C = N) 1625m,1603 cm^−1^. ^1^H NMR (CDCl3, δ ppm, J Hz): 8.0, 7.85 [s, 2H, HC=N], 6.6 [s, 1H, H2], 6.2 [d, 1H, H4, J(HH) = 7.7], 6.0 [dd, 1H, H5, J(PH) = 7.4], 1.43 [s, 6H, Me, J(PtH) = 67.4]. ^31^P-{^1^H} NMR (CDCl_3_, δ ppm) 109.3 [t, 1P, J(PP) = 23.0 Hz], 92.6 [d, 2P], 18.0 [s, 2P, J(PtP) = 2426].

### 3.3. Crystal Structure Analysis and Details on Data Collection and Refinement

Compound **5b**: 2215129, compound **10**: 2215130 (starting from 3a) and 2144107 (starting from 2a) CCDC and CCDC. See Appendix A.

## 4. Conclusions

In conclusion, the palladacycles bearing 1,3- and 1,4-bidentate imines are appropriate starting materials to prepare new metaloligands, especially those derived from the meta-substituted ligands. The metallacycles undergo stepwise cleavage of the C=N double bonds to render the two non-coordinated formyls groups; treatment with aminophosphine ligands gives species with two phosphorus donors. The resulting complexes, which may be coined as bidentate [P,P] metaloligands, can effectively bind to palladium or platinum in a chelate fashion to render stable homo- or hetero-double nuclear compounds. In the course of this research paper, we have found a totally serendipitous result, which shows the ease with which bonds on palladium can self-modify in palladacycles. Namely, compounds **2a** and **3a** are spontaneously and independently transformed into the double nuclear complex **10**, inclusive of a second metalation of the phenyl ring. Whether or not the tentatively proposed mechanism for this situation is totally or in part correct, the X-ray diffraction result is unquestionable evidence of this outcome. A further interesting feature of **10** is the value for the palladium-nitrogen bond, which makes it the shortest distance found to date in this type of complexes. Further studies are presently in progress in order to shed more light regarding this process.

## Data Availability

The data is only available form the authors on request.

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
