# Peer review of "An Innovative Structural Rearrangement in Imine Palladacycle Metaloligand Chemistry: From Single-Nuclear to Double-Nuclear Pseudo-Pentacoordinated Complexes"

_molecules, 2023, doi:10.3390/molecules28052328_

Round 1

Reviewer 1 Report

The authors reported “An innovative structural rearrangement in imine palladacycle 2 metalloligand chemistry: from single-nuclear to double-nuclear 3 pseudo-pentacoordinated complexes”. I believe that the manuscript can be published in Molecules after minor revisions.

The work is well articulated and supported by relevant experimental data. The characterizations of the products obtained are adequate.

 I believe that the abstract, as it has been set up, is not adeguate. It is not only possible to report the labels of the molecules (for example 1a, 3b) but it is necessary for each chemical compound, mentioned in the abstract, to report the name (complete or indicative) of the compound itself.

I also believe that the abstract should be enriched.

Before the manuscript can be accepted, the authors must make the required corrections.

Author Response

Reviewer 1 Reply

The names for the compounds have been included.

The Abstract has been amended and adequately enriched.

Reviewer 2 Report

The manuscript is in general acceptable.

Two crystal structures are mentioned. At least the crystal data and the R factors must be mentioned in the main text.

In this regard the crystal structure diagrams must show the relevant hydrogen atoms.

Author Response

Reviewer 2 Reply

Selected crystallographic data, inclusive of the R factors, is included in the legends for the structures of 5b and 10.

The crystal structures how the relevant hydrogen atoms, namely those for the metalated phenyl ring and the C=N and C=O groups.

Reviewer 3 Report

The manuscript by al Janabi et al. is devoted to the study of the chemistry of cyclometallated palladium complexes with triphos ligand. The authors obtained interesting results and the manuscript is certainly of interest for publication in Molecules. However, as the authors note, the purpose of the study has undergone certain changes in the course of the work, which, in my opinion, also affected the structure of the manuscript.

The abstract is completely uninformative because it cannot be understood without Scheme 2. In Scheme 2 itself, there are no conditions for the transformations of iii and vii. The numbers of compounds in the text of the manuscript appear in the most unpredictable way - first 5b and 10, then 2a and 2b, 3a and 3b. With neither 1 nor 4a are simply not mentioned. All this makes the manuscript difficult to read - it feels like when you read, you are wading through the jungle. I would strongly recommend splitting Scheme 2 into two schemes describing compounds with indexes a and b separately, while swapping them so that complex 10 is discussed at the end. This would give harmony to the entire manuscript.

A few remarks about the mechanism shown in Scheme 3. Obviously, for the formation of complex 10, at least half of the initial complex 2a must be destroyed in order to give the required amount of cyclohexylamine and the Pd(triphos) fragment. It is clear that the imine 2a1 must first be formed, and then the imino group must direct the CH activation of the aromatic ring by the palladium fragment. However, the formation of the 2a2 complex arises only as a bold assumption. If the authors had first discussed the line of compounds with index b, in which complex 5b had been isolated and characterized, then Scheme 3 would have looked much more logical.

The language of the manuscript is also in some places not very clear due to the use of very long and complex sentences (eg. lines 23-28).

What is reference 2? This is a book? Reference 21 is missing the title of the article.

Author Response

Reviewer 3 Reply

-The manuscript has been reorganized following Reviewer´s 3 suggestions.

-Accordingly, first the a compounds are described. Then compounds of type b. Lastly, the final part of the Results and Discussion section focuses on the descripton of compound 10, its crystallographic study and the possible mechanism of formation.

-The language has been checked. Lines 23-28, presently 31-38 have been corrected.

-Reference 2 is a book.

-The title missing in reference 21 has now been included.

Round 2

Reviewer 3 Report

The authors have done a good job modifying the manuscript and now it looks well structured and easy to read. I am happy to recommend it for publication in its present form.